# A Comparative Evaluation of Cement and By-Product Petrit T in Soil Stabilization

**Wathiq Al-Jabban** [1,2] , **Jan Laue** [1] , **Sven Knutsson** [1] **and Nadhir Al-Ansari** [1,*]

1 Civil, Environmental and Natural Resources Engineering, Lulea University of Technology, 971 87 Lulea, Sweden; wathiq.al-jabban@ltu.se or wathikjasim@yahoo.com (W.A.-J.); jan.laue@ltu.se (J.L.); sven.knutsson@ltu.se (S.K.)
2 Civil Engineering Department, University of Babylon, P.O. Box 4 Hilla, Babylon 51001, Iraq
* Correspondence: nadhir.alansari@ltu.se; Tel.: +46-725390767 or +46-920491858

**Abstract:** This study presents a comparison between the effectiveness of adding low binder amounts of industrial by-product Petrit T as well as cement to modify and improve fine-grained soil. Binder amount was added by soil dry weight; cement at 1%, 2%, 4% and 7% and Petrit T at 2%, 4% and 7%. The unconfined compressive strength (UCS) was used as an indicator of soil strength. In addition, the consistency limits, laser particle size analysis, and pH tests were also conducted on the treated soil. The samples were cured at 20 °C for different periods from 7 to 90 days before testing. Results indicate that cement is more effective at improving the physical and engineering properties of the treated soil. Soil plasticity index decreases after treatment and with time. Liquidity index and the water content to plastic limit ratio are introduced as new indices to define the improvement in the workability of treated soil. Soil particle size distribution is changed by reducing the clay size fraction and increasing the silt size fraction after treatment. The findings confirm that adding small binder contents improve soil properties, which subsequently reduce the environmental threats and costs that are associated with using a high amount of binder.

**Keywords:** Petrit T; cement; secant modulus; workability; soil strength; solidification; pH

---

## 1. Introduction

Soil improvement by adding stabilizing agents is the most common, effective and economical technique to enhance the strength characteristics of soft soil [1]. A wide range of soils can be improved by adding different types of hydraulic binders, but the cement has been the most popular and successful stabilizer used [2]. Nowadays, the use of waste materials from the various industrial process as a cementitious binder has increased to stabilize different types of soils. These by-product materials are considered to be easily available and cheap compared to the traditional binders [3–5]. In addition, there is an environmental benefit from the reuse of these types of by-products. Its contribution is to decrease environmental impact posted by producing these material [6].

Generally, cement improves the physical and mechanical properties of soils by produce primary and secondary cementitious materials [7]. After adding cement to wet soil, three types of primary cementing components are produced as a result of the hydration reaction of cement: calcium-silicate hydrate (CSH), calcium-aluminate-hydrate (CAH), and hydrated lime (Ca(OH)$_2$). Here, C (CaO), S (SiO$_2$), A (Al$_2$O$_3$), and H (H$_2$O) are employed. A pozzolanic reaction produces a secondary cementitious material due to the reaction between the hydrated lime and silica and alumina from clay minerals and provides further cementing components of CSH and CAH [8–10].

The benefit of using cement in the soil stabilization field has been extensively investigated [11–18]. In addition, the possibility of using industrial by-product materials to improve soft soils has been



encouraged by many studies [6,19–23]. Enhanced soil strengths, reduced soil plasticity and swelling potential are the most desirable outcomes from the treatment. For certain soil stabilization applications, the cost of the binder itself could result in a considerable saving in the overall costs of the project [24]. Petrit T is a waste product from sponge iron production [18]. It is considered a cheap by-product material [25]. Therefore, in order to decrease the environmental impact and costs, finding new binders as alternatives to Portland cement is important. Thus, there is a need to compare the effectiveness of using cement and by-product Petrit T as stabilizing agents.

This paper presents a comparative evaluation between using Portland cement and by product Petrit T to enhance the physical and mechanical properties of sandy clayey silt soil. An extensive experimental program was carried out in this study, including tests of unconfined compressive strength, laser particle size, consistency limits, and pH tests, using various binder amounts and curing periods.

## 2. Materials and Methods

The samples used in this study were prepared from a natural soil from Gothenburg, south west coast of Sweden. The soil consists of 55% silt, 29% fine sand and 16% clay. The natural water content is 30%, the liquid limit 37% and the plastic limit 20%. The optimal moisture content is 12%, and the proctor density is 1.97 t/m$^3$. The ignition test showed that the soil has 4% organic content. Therefore, according to the Swedish standard, natural soil is classified as sandy clayey silt (saclSi) with low organic content [26–28].

Natural soil was firstly homogenized and then mixed with Petrit T as well as the cement in various percentage from 1 to 7% as dry mass of soil and cured for 7, 14, 28, 60 and 90 days before testing.

Unconfined compressive tests (UCS) were used as an indicator to predict the soil strength enhancement before and after treatment. A series of consistency limits tests were performed on the treated soil after various curing times to investigate the effect of binder types and amounts on the soil plasticity index. Laser particle size analyzer tests were conducted to study the effects of binder types on the particle size distribution (PSD) of the soil before and after treatment. These were measured by using CILAS 1064 laser particle size analyzer in the liquid mode with a measurement range from 0.04 to 500 μm. The pH tests were used as an indicator of soil-binder reaction by measuring the pH value of the soil after treated with a binder and cured for various curing periods.

The specimens for particle size distribution tests (PSD) were prepared and cured in an identical way to the UCS specimens. Binder content was added as dry material with 4% and 7% by dry weight of soil. After 28 days of curing time, the specimens were pressed out of the tubes with the help of a mechanical jack directly before testing. Prior to the PSD test, both stabilized and unstabilized soil specimens were air-dried and ground through 0.5–0.063 mm sieves. Then, about 5 g of that passing from 0.063 mm sieve was mixed with sufficient distilled water and subjected to the PSD analyzer (CILAS 1064).

The soil and the binders as well as the specimen preparation and testing methodology are described in more detail in [18,29,30].

## 3. Results and Discussion

### 3.1. Consistency Limits (Atterberg Limits)

Figure 1 shows the immediate effect of adding various amounts of binder (cement or Petrit T) on the soil consistency limits. The addition of low amounts of either type of binder (1 to 4%) had an immediate effect on increases in both the liquid limit (LL) and the plastic limit (PL). The liquid limit remained almost constant for further increases from 4 to 7%, although it dropped at higher percentages of binder amount (10% and 15%). After treatment, the plastic limit significantly increased at low binder contents (1 to 7%) compared to a slight increase when the amount of binder was increased between 7 to 15%. Consequently at low amounts of binder (1–4%), the plasticity index (PI) slightly increased and then decreased as the amount of binder was increased. From Figure 1, it can be seen that the trends

have generally been in the same direction, but with more marked effects for the cement treatment on both liquid and plastic limits. Moreover, both binder types have approximately the same behavior trend on the reduced plasticity index for up to 10% of binder content. In contrast, cement has more effect on reducing the plasticity index at high binder content (15%).

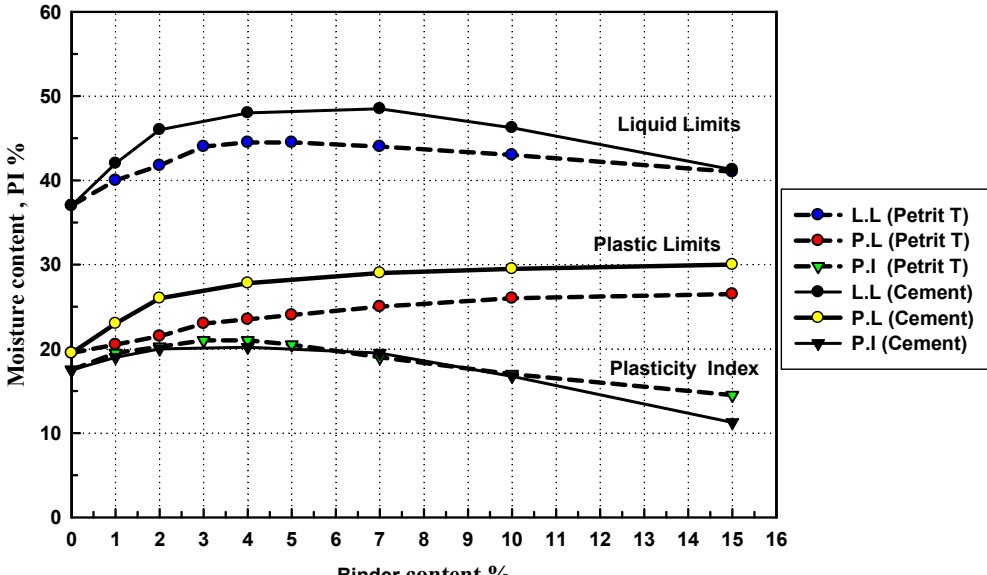

**Figure 1.** Consistency limits versus binder content of soil after one hour of treatment.

The immediate increase in the liquid limit and plastic limit was attributed to the flocculation and agglomeration of soil particles caused by the hydration reaction of the binder. A similar trend is found in previous studies. For cement treatment, reference [31] suggested that the presence of entrapped water within the intra-aggregate pores due to flocculation and agglomeration of soil particles was the main reason behind an increase in the liquid limit after treatment. In contrast, increased amounts of binder produced an increase in cementitious materials and this lead to decreased liquid limits. References [32–34] have reported about increases in the liquid and plastic limit of soils after have been treated with various binder types.

Flocculation and agglomeration of the soil particles after treatment is the main reason behind the immediate change in the liquid and plastic limit of soil, which leads to a reduction in the plasticity index of soil with the increasing amount of binder. A similar trend is observed for a wide range of soil treated with different binder types [35–39].

Figure 2 shows the curing time effects on the plasticity index after treatment. It can be seen that the trends are generally in the same direction (a decrease in the plasticity over time), but with more marked effects for the cement treatment. The reduction in the plasticity index is increased as binder content increased. Moreover, Figure 2 shows that a 2% of Petrit T has approximately the same effects as adding 1% of cement. For both binder types, the decrease in the plasticity index is mainly due to increases in plastic limits and decreases in liquid limits over time. A similar trend has been observed in previous studies [22,31,32].

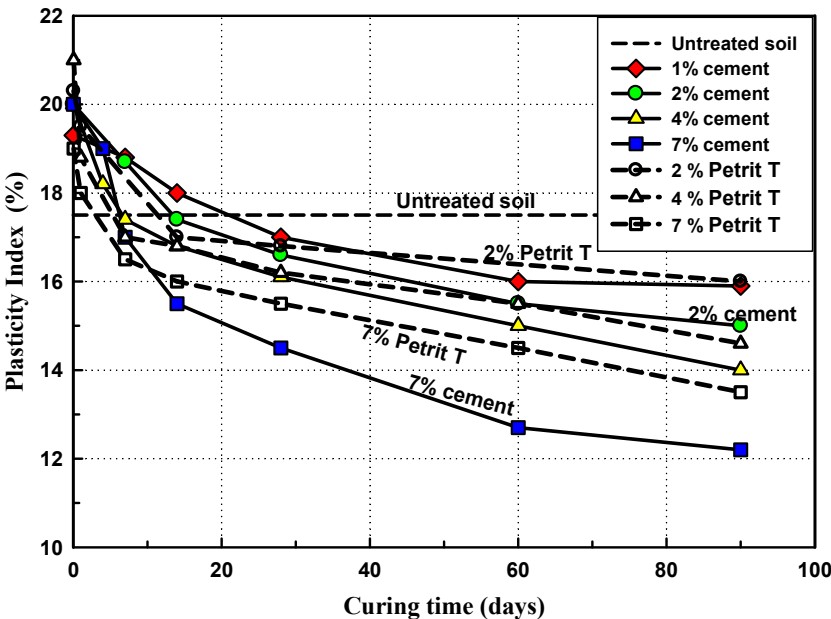

**Figure 2.** Plasticity index versus curing time.

### 3.2. Water Content and Density

Figure 3 shows the immediate effects of adding Petrit T and cement on the soil water content. It is seen that the addition of both binders has the same trend of reducing the soil water content from its initial value (30%), but the effect is more marked for the cement rather than the Petrit T. The reduction in moisture content of soil is mainly related to introducing dry solid particles into the soil, as well as the hydration reaction between the binder and water. Cement has a greater effect on reducing the initial water content compared to Petrit T, and this is particularly so when binder content levels exceed 7% (see Figure 3). This can be attributed to the variation in hydraulic properties between the two binders, which relates to the ability of each binder type to reacting with water. A similar observation has been made in previous studies [39–41].

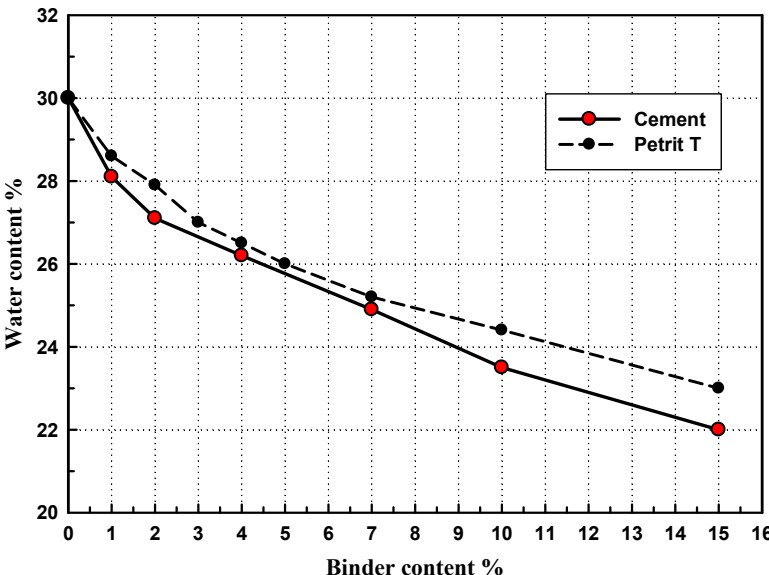

**Figure 3.** Water content of soil versus binder amount after one hour of treatment.

The curing time effects on the soil moisture content of the treated soil with various amounts of the binder is presented in Figure 4. For both binder types, the soil moisture content is further

decreased with time, with a more dominated effect for cement treatment during the first 28 days of curing. In contrast, a gradual decrease in soil water content with curing time is observed for Petrit T treatment. This can be attributed to the lower reactivity of the main component $C_2S$ (belite) in Petrit T (compared to cement), which has four major components: $C_3S$ (alite) and $C_2S$ (belite) in addition to $C_3A$ (aluminate) and $C_4AF$ (Ferrit). The further reduction in soil moisture content is attributed to the hydration and pozzolanic reactions of binders as the samples were cured in a semi-sealed condition. Small leaks in the plastic caps of the UCS samples can be the reason behind the increase in water content for treated soil with 4% and 7% cement content as shown in Figure 4.

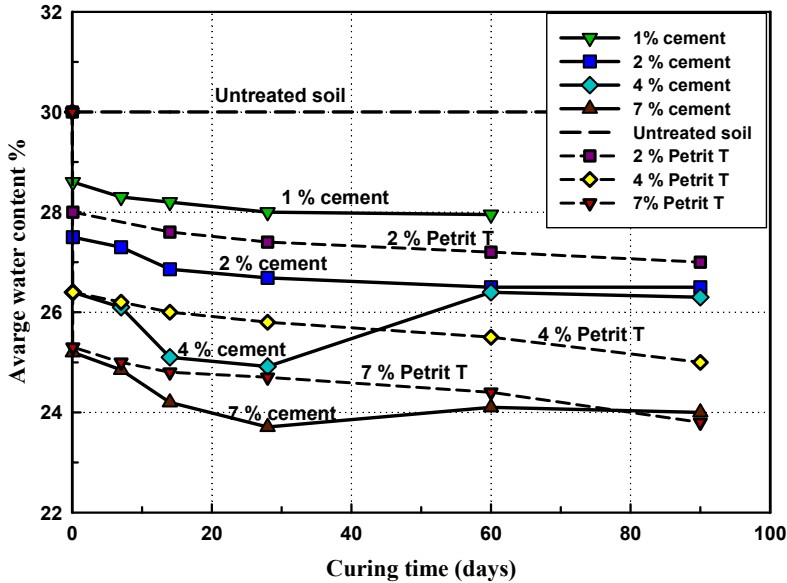

**Figure 4.** Soil water content versus curing time and binder type and content.

From Figures 1–4, it can be seen that change the soil consistency limits after treatment is accompanied by a reduction in soil moisture content. Workable soil is a term used to explain how the soil can be easily excavated, loaded, transported, compacted, etc. [18]. The immediate reduction in soil moisture content and plasticity index transfer the soil from being plastic to more granular soil, which facilitates higher workability. As pointed out by [42–44], the workability of clay has been shown to improve with a reduction in the plasticity index. The majority of researchers compared the improvement in workability of treated soil by reducing the plasticity index after treatment e.g., [17,39,45]. The immediate increase in the plasticity index combined with the initial reduction in soil water content due to adding small amounts of binders doesn't mean that the workability of soil did not improve. For this reason, using the plasticity index as an indicator to measure the improvement in workability of the treated soils is of limited reliability. Therefore, the liquidity index (LI), as expressed in Equation (1), can be used to explain the affinity between the consistency limits and moisture content of treated soil with two binder types (cement or Petrit T). For the immediate effect (one hour after treatment), Figure 5 shows the relation between the liquidity index (LI) versus binder content. It is noticed that the liquidity index (LI) reduced from 0.6 (soil without binder) until it becomes very close to the plastic limit (LI = 0) at 3% and 7% of cement and Petrit T content, respectively. As the binder amount is increased, further reduction in the liquidity index is observed below the plastic limit.

$$LI = \frac{(W_c - PL)}{PI}, \tag{1}$$

where: LI: liquidity index, PI: plasticity index, PL: plastic limit and $W_c$: soil water content.

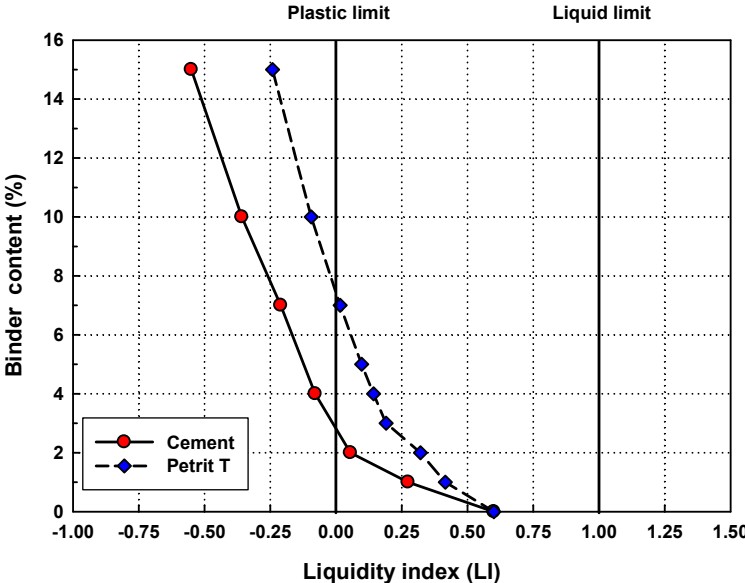

**Figure 5.** Binder content versus liquidity index of soil after one hour of treatment.

For further investigation of the curing time effect on the affinity between soil moisture content and the liquidity index, it is observed that the increase in curing time cause a further decrease in the liquidity index and the soil moisture content as illustrated in Figure 6. Thus, adding cementitious binder (cement or Petrit T) results in a reduction in the water content towards the plastic limit of soil after treatment. As mentioned by [32], an increase in soil strength is observed when the soil water content is reduced from the liquid limit to the plastic limit. Consequently, the relationship between the consistency limits and soil strength after treatment will be presented and discussed in more detail later in this study.

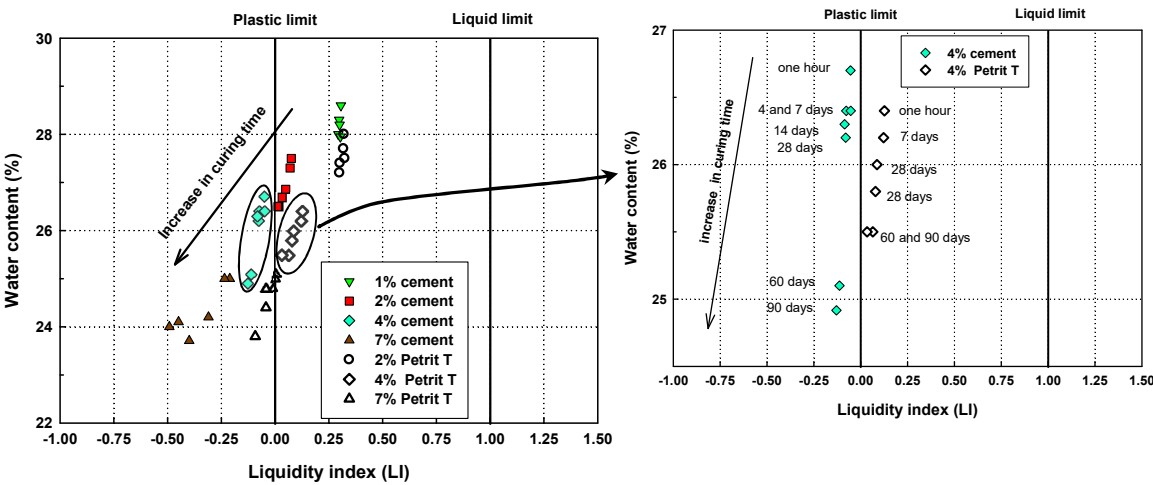

**Figure 6.** Liquidity index versus soil water content for all curing times.

The effect of the added cement and Petrit T on the bulk density of the soil after being treated with various percentages of binder is presented in Figure 7. For both binder types, the soil density increases as the binder content and curing time is increased. The bulk density of the cement specimens is higher than for the Petrit T specimens due to adding 1 to 4% binder content. In contrast, a slight increase in the specimen density of Petrit T compared to cement is observed at 7% binder content and longer curing period (see Figure 7). Consuming water content due to hydration and pozzolanic reactions produces a large amount of solid particles in the soil, which consequently leads to an increase in the

density of the soil after treatment. The increase in specimen's density is attributed to the production and deposition of CSH and CAH, which subsequently fill the pore voids [13,31,32,46].

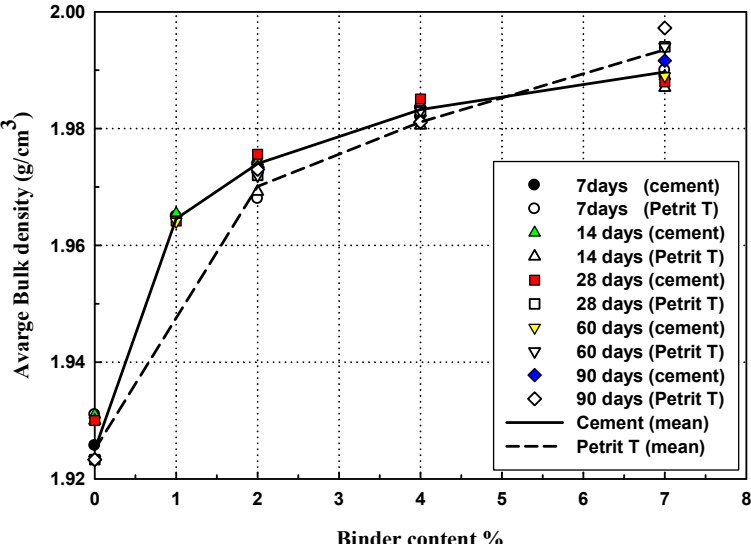

**Figure 7.** Specimen density versus binder content.

### 3.3. Particle Size Distribution (PSD)

Table 1 present the results of the laser PSD analysis for natural soil and treated soil with cement and Petrit T after 28 days of curing.

**Table 1.** Percentage of clay and silt size fractions of untreated and treated soil measured by the CILAS 1064 laser particle size analyzer.

| Soil | Binder Amounts % | Clay-Sized Particle % (<2 μm) | Silt-Sized Particle % (2 to 63 μm) |
|---|---|---|---|
| Soil without treatment | 0 | 17 | 83 |
| Cement treated soil | 4 | 13 | 87 |
| | 7 | 10 | 89 |
| Petrit T treated soil | 4 | 14 | 86 |
| | 7 | 13.4 | 86.6 |

For both binder types, the general tendency is that the percentage of the clay size particles is reduced while the percentage of silt-sized particles is increased after treatment. Cement has more effect compared to Petrit T by lowering PSD curves towards the granular side. The changes in the PSD after treatment is attributed to flocculation and agglomeration of the fine particles during a short time after treatment. This is in addition to the pozzolanic reactions as a long-term effect [22].

Flocculation leads to agglomerate the fine particles during a short period. Whilst over time, the pozzolanic reaction produces more cementitious materials of CSH and CSAH which coating the surface of soil particles. Both reactions are contributed to increasing the fraction of coarse-grained particles after treatment [47,48]. Similar trends in changing the PSD are consistent with previous studies [22,31,49].

### 3.4. pH Value

Figure 8 shows the pH value of treated soil after one hour of adding cement or Petrit T. For both binders, the soil pH value rose from 5 to about 12.35 as the binder content was increased up to about

7%. The pH of the soil slightly increases to 13 with an increase in the amount of binder to 15%. It is observed that both binder types have the same initial effect on increasing pH after treatment. The reaction of both binders with water leads to the release of calcium ions ($Ca^{2+}$) on the surface of soil particles and an increase in the pH value [31].

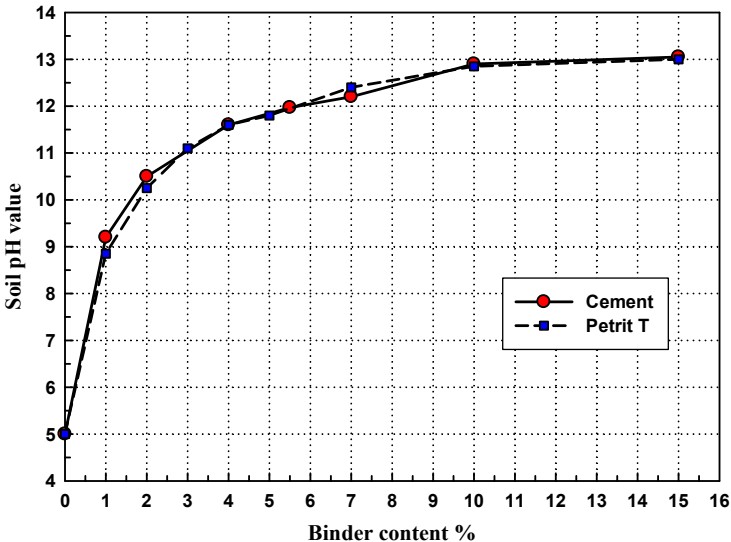

**Figure 8.** The pH value of soil after one hour of treatment versus binder content.

The curing time effects on the pH value of the soil is illustrated in Figure 9 for both cement and Petrit T treatments. For both binder types, the general trends are that the pH value gradually decreases as curing time is increased regardless of binder content. The major decreases in pH value for the cement treatment occurred during the first 28 days. In contrast, Petrit T has a gradual decrease over time. The consumption of ($OH^-$) is the reason behind the decrease in pH value [50]. The reduction in pH value for cement treatment is higher than Petrit T. This can be attributed to the production of more CSH and CAH components from hydration and pozzolanic reactions. A similar trend for decreasing the soil pH value with curing time was investigated by [51].

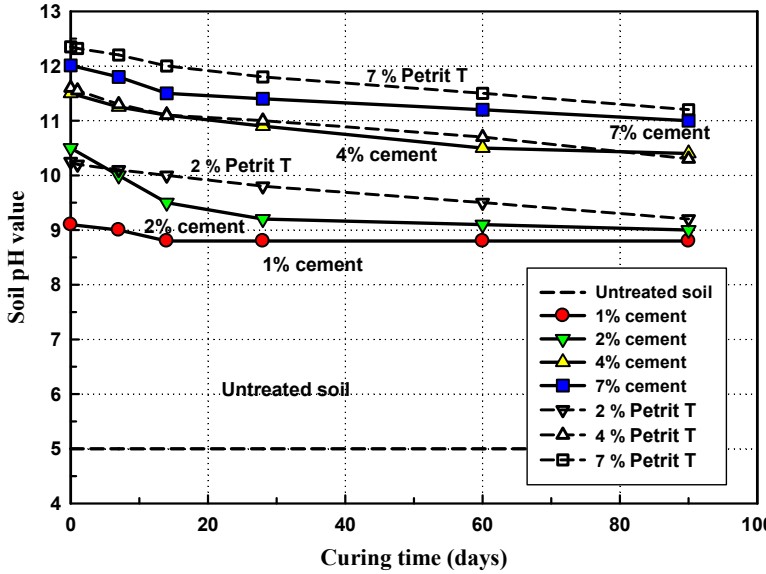

**Figure 9.** The pH value of treated soil versus curing time and binder amount.

### 3.5. Soil Strength

The unconfined compressive strength ($q_u$) was used as an indicator to investigate the enhancement in soil strength. These tests were conducted on the untreated and treated soil, both of which were prepared in a similar way, for the same curing period and conditions. For the untreated soil, Figure 10a shows almost no increases in the unconfined compressive strength ($q_u$) over time. Figure 10b–d show the effects of adding the two binder types (cement and Petrit T) on the improvement in soil strength.

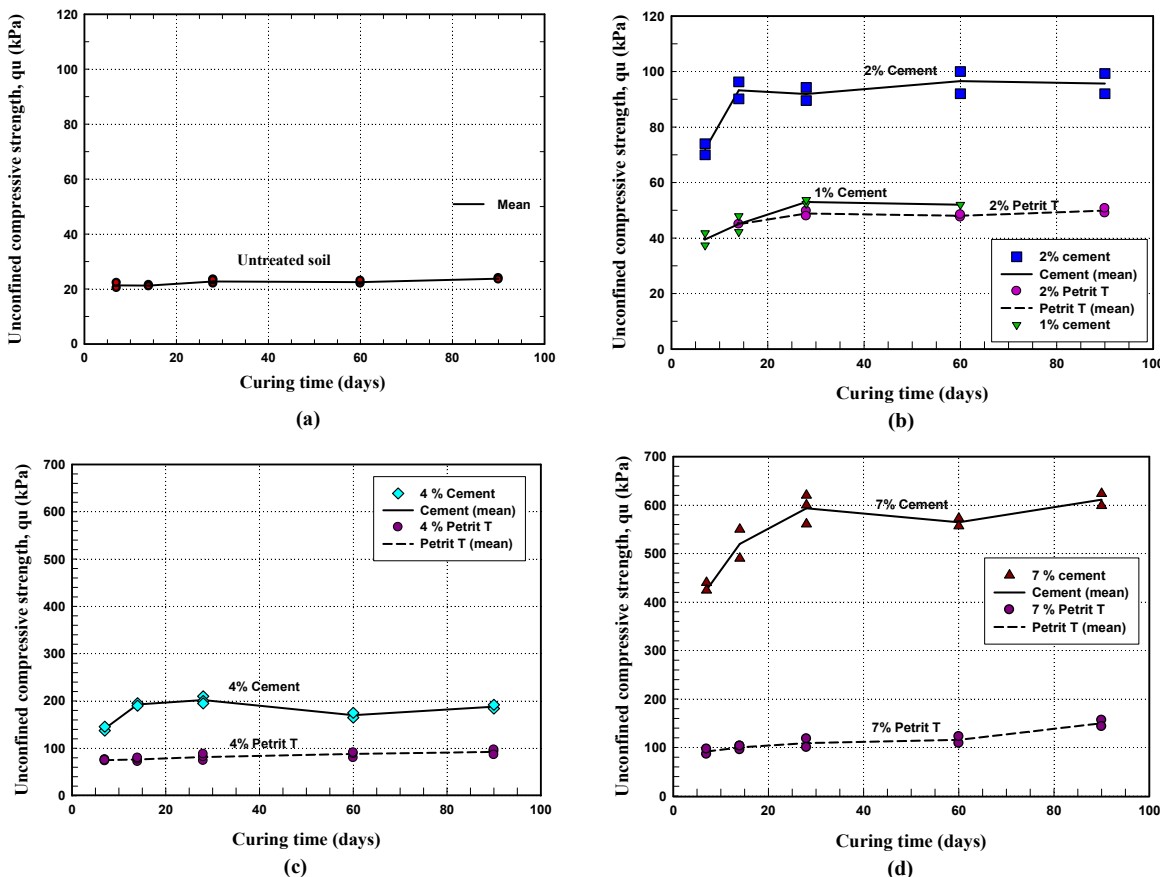

**Figure 10.** Unconfined compressive strength (UCS) versus curing time for (**a**) Untreated soil; (**b**) Treated soil with 1 and 2% cement and 2% Petrit T; (**c**) Treated soil with 4% of cement and Petrit T; and (**d**) Treated soil with 7% of cement and Petrit T.

Cement has four major clinker minerals, $C_3S$ (alite), $C_2S$ (belite), $C_3A$ (aluminate) and $C_4AF$ (Ferrit) [50]. The reactions of $C_3S$ and $C_2S$ have been the main contributors to the gain in strength, whilst the reactions of $C_3A$ and $C_4AF$ have made only a minor contribution. The reaction of $C_3S$ produced rapid hardening of cement during a short curing time. Whilst the reaction of $C_2S$ has a lower reactivity, therefore, the gain in the strength is observed after a long curing period.

For the by-product Petrit T, the X-ray diffraction (XRD) (provided by the manufacturer) shows that Petrit T consists mainly of 57% dicalcium silicate (similar to the clinker mineral $C_2S$ in cement) 28.3% calcium silicon-aluminate, 11.5% silicon dioxide and 3% portlandite. The mineral part represents 74% of the whole mass, the rest consists of amorph carbon (21%) and iron (5%). The major component mainly of Petrit T is the dicalcium silicate and calcium-silicon-aluminate. Therefore, the improvement in soil strength over time can be explained by the slower reaction of $C_2S$.

An improvement in soil strength after treatment for both cement and Petrit T was observed by increasing in the amount of binder and curing time. This gives an indication of generating new components of CSH and CASH as a result of hydration and pozzolanic reactions.

At 1% and 2% binder content, the strength of soil increased to 50 kPa due to the addition of 2% of Petrit T, and 52 and 97 kPa due to the addition of 1 and 2% of cement content, respectively. For both binders, the main improvement in strength occurred during the first 28 days of curing. However, the long curing time at 60 and 90 days showed no further improvement in soil strength (see Figure 10b). The lack of soil strength improvement at the longer curing times can be explained by the pH value. The pH value are 9.2 and 9 for the 60 and 90 days of curing respectively for both cement and Petrit T (see Figure 9). As observed by several researchers [51–53], a soil–binder mixture with a pH value higher than 10 is enough to initiate the pozzolanic reaction by dissolving the silicates and aluminate and produce additional cementitious materials of CSH and CAH. The pH value at 60 and 90 days is lower than 10, therefore no additional effect for the long curing period on further improvement in soil strength.

A gradual increase in strength of the soil is observed with curing time when the binder content was increased from 2 to 4% for cement and Petrit T (see Figure 10c). Similar trends were noticed when the amount of binder is increased from 4 to 7%, with notable strength development for cement during the first 28 days of curing compared to a gradual increase in soil strength over time for Petrit T as shown in Figure 10d. For the soil specimens treated with 4% cement content and cured for 60 and 90 days, the reduction in soil strength can be attributed to an increase in water content as discussed earlier (see Figure 4). A similar trend of reduced soil strength for low cement content under saturation conditions has been observed by [11,39].

The soil strength increased with the amount of binder and curing time due to hydration and pozzolanic reactions. For cement treatment, the soil strength improvement was noticeable during the first 28 days of curing, while it was more pronounced for Petrit T at longer curing periods (90 days). This trend is consistent with previous studies [14,15,22,23,35].

### 3.6. Consistency Limit—Unconfined Compressive Strength (UCS) Relationship

Figure 11 shows the relationship between the liquidity index (LI) and the unconfined compressive strength. It can be seen that there is a linear relationship between the increases in the unconfined compressive strength and decreases in the liquidity index. The treated soil was within the plastic range (LI < 0) when the soil strength was less than 100 kPa.

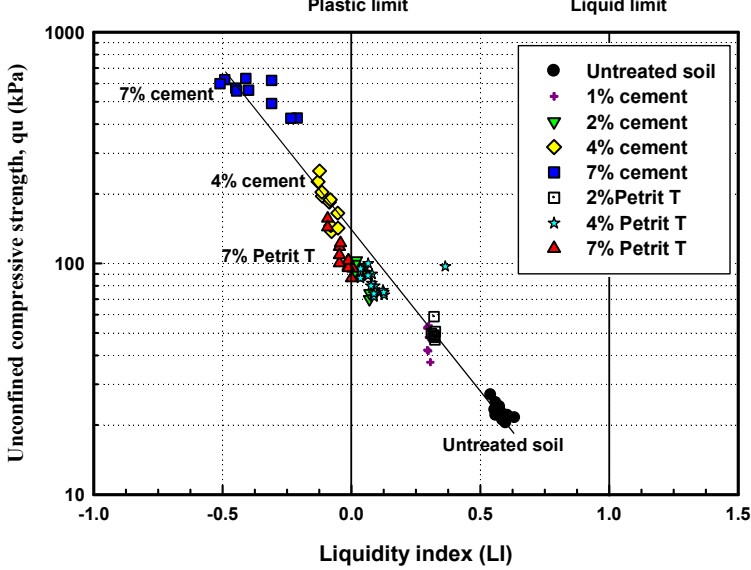

**Figure 11.** UCS versus liquidity index for all tests.

As discussed earlier, the addition of binders had the effect of reducing the water content towered the plastic limit of soil (see Figures 5 and 6). For further investigation about the relations between consistency limits and the UCS, a water content to the plastic limit ratio $\left(\frac{\omega}{PL}\right)$ can have a major effect as presented in Figure 12. As the water to plastic limit ratio decreased, an increase in the unconfined compressive strength was observed. Moreover, a more scattered pattern was noticed at the water/plastic limit ratio lower than 1 (see Figure 12), and the scattering is decreased with increasing the ratio to more than 1. This can be attributed to more difficulties in finding the plastic limit of treated soil at these level of soil strength. The treated soil occurs within the plastic range when the soil strength is less than 100 kPa. A similar trend is consistent with [32] for soil treated with various binder types.

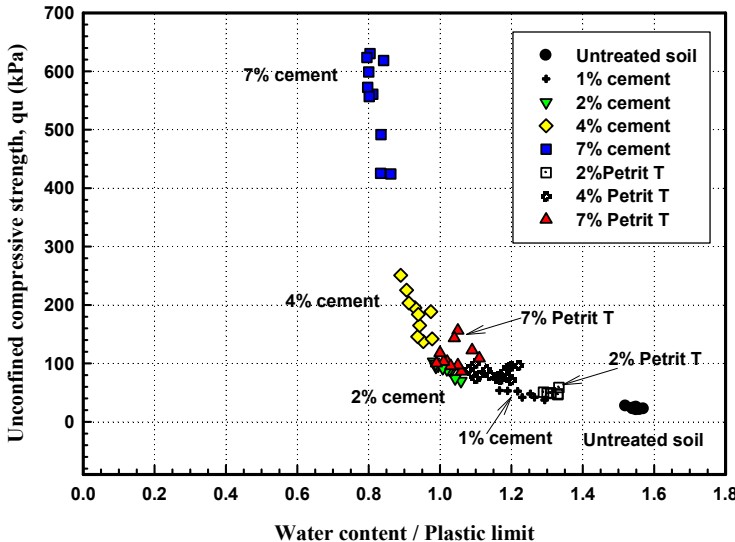

**Figure 12.** UCS versus water/plastic limit ratio for all tests.

From the above, it can be concluded that the liquidity index (LI) and $\left(\frac{\omega}{PL}\right)$ ratio can be used to explain the relationships between various soil properties and the workability of soil after treatment. The workability improved immediately after treatment with the reduction in the liquidity index. Continuous improvement in soil workability was observed over time due to further reduction in the liquidity index within the plastic range when the LI > 0 or when the water/plastic limit ratio was greater than 1.

*3.7. Stress-Strain Behaviors*

For two binder types, Figure 13 shows typical stress-strain curves at different binder content and curing times. Figure 13 shows that the soil without any treatment has a 24% failure strain in addition to the low peak stress of 23 kPa. For the treated soil with cement or Petrit T, the general trends are that the peak strength of treated soil increases with an increase in binder content. Whilst, failure strain, corresponding to the peak stress, decrease as the amount of binder is increased. During the UCS tests, small cracks clearly observed on the surface of the samples when the stress reaches the peak point.

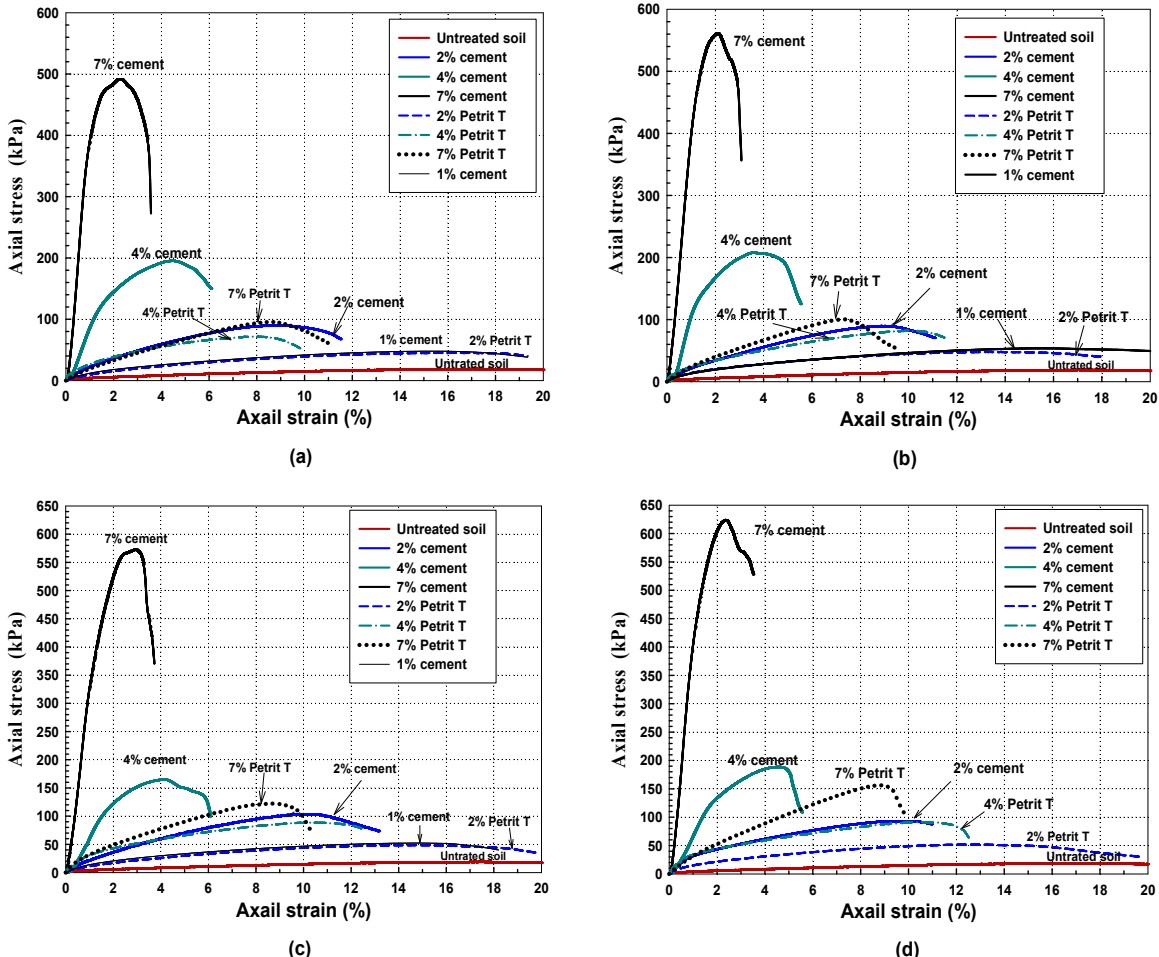

**Figure 13.** Stress–strain curves for the untreated and treated soil with different binder content after (**a**) 14 days curing time; (**b**) 28 days curing time; (**c**) 60 days curing time; and (**d**) 90 days curing time.

For the cement treatment, a significant improvement in the stress-strain behaviors was observed during the first 28 days of curing, with small changes at longer curing times. In contrast, for Petrit T, the most important change occured with the long curing period (90 days) (Figure 13b–d). It can be seen that the stress-strain curve for 4% Petrit T improved with a curing time from 14 days (Figure 13a) until it was almost similar to the stress-strain curves of 2% cement at 90 days curing time (see Figure 13d). A similar observation was valid for the addition of 2% of Petrit T compared to 1% of cement. The soil specimens treated with 7% cement content exhibit a more brittle failure compared to the lower cement content and Petrit T specimens, which showed more ductile behaviors (Figure 13d). Thus the failure mode gradually changed from plastic to brittle failure with an increase in the binder content.

A similar trend for a change in the behavior of the stress–strain curves has been observed by other researchers for various binder types [14,15,31,39].

### 3.8. Strain at Failure

Figure 14 shows the axial failure strain versus curing time for untreated and treated soil. For the soil without any treatment, high failure strain is observed (Figure 14a). In contrast, due to the addition of different binder content of cement or Petrit T, failure strain is significantly reduced with more effect from the cement treatment, as shown in Figure 14b–d. Petrit T, compared to cement, exhibits major decreases in failure strain during the first 28 days. This can be attributed to produce more cementing components from the cement than Petrit T. Generally, the average failure strain for the long curing period (90 days) and 7% binder content was less than for shorter curing times (seven days).

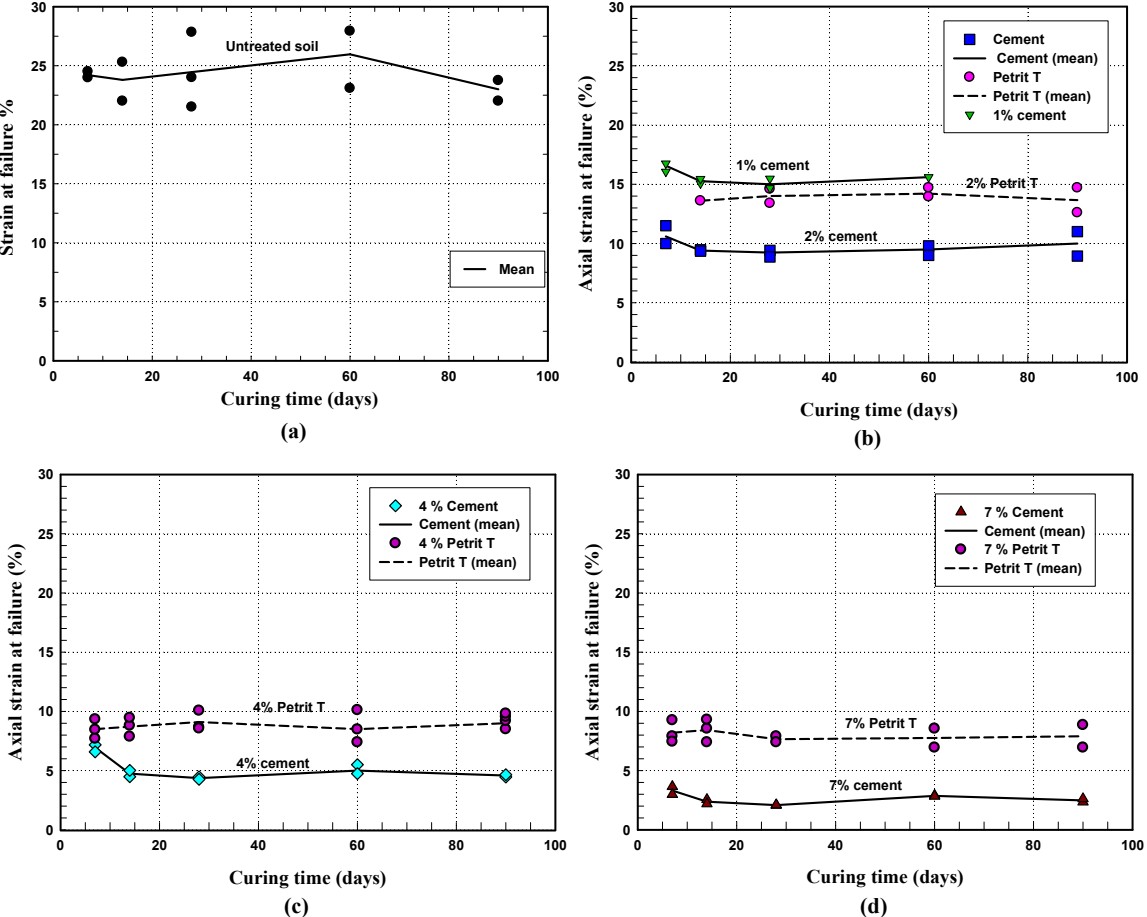

**Figure 14.** Strain at failure versus curing time for (**a**) untreated soil; (**b**) treated soil with 1 and 2% cement and 2% Petrit T; (**c**) treated soil with 4% of cement and Petrit T and (**d**) treated soil with 7% of cement and Petrit T.

Figure 15 shows the axial failure strain versus the UCS for both cement and Petrit treatment and all curing times. It is observed that the strength of soil increased, whilst failure strain decreased as the amount of binder increased. A significant reduction in the axial failure strain was noticed regarding different cement and Petrit T treatments, which led to an increase in strength of treated soil to about 150 to 200 kPa. Further increases in soil strength proved less significant on decreasing failure strain. The result is consisted with [15,32,54].

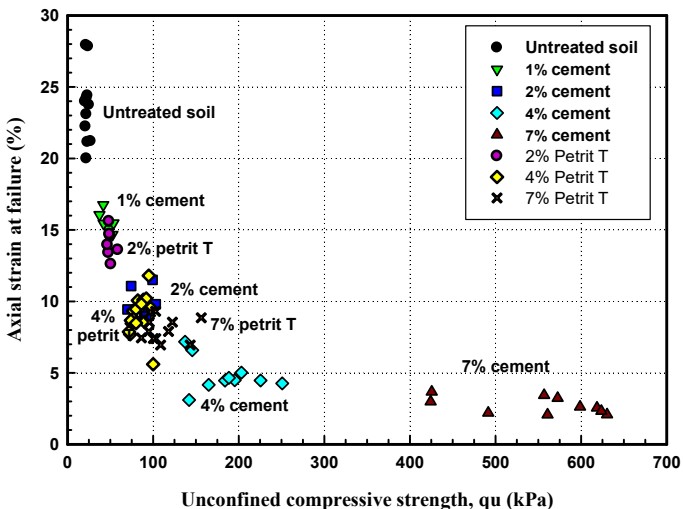

**Figure 15.** Axail failure strain versus UCS strength for all tests.

## 3.9. Soil Stiffness

The effects of the two binder types (cement and Petrit T) on the stiffness of treated soil with various binder content and curing times is presented in Figure 16. The stiffness in terms of $E_{50}$ was determined as a secant modulus of elasticity from the stress-strain curves at 50% of the maximum unconfined compressive strength ($q_u$). It can be seen that the stiffness of the treated soil increased as the binder content and curing time increased (see Figure 16).

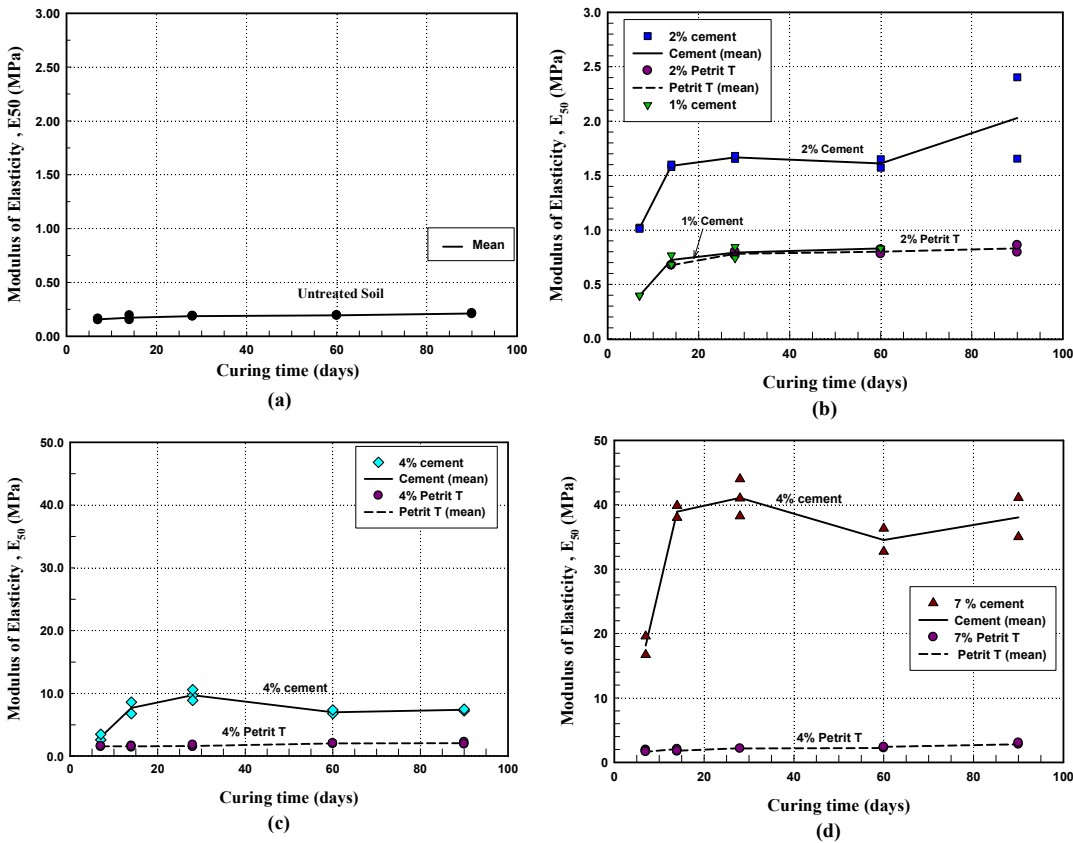

**Figure 16.** Modulus of elasticity versus curing time and binder content for (**a**) untreated soil; (**b**) treated soil with 1 and 2% cement and 2% Petrit T; (**c**) treated soil with 4% of cement and Petrit T and (**d**) treated soil with 7% of cement and Petrit T.

The development characteristic curve of the modulus of elasticity ($E_{50}$) over time has very similar trends to the development of unconfined compressive strength as presented in Figure 10. This can be attributed to generating new cementing components of CSH and CAH as a result of hydration and pozzolanic reactions. A higher binder content produce more cementitious materials of CSH and CAH and vice versa. As mentioned earlier, the production and deposition of cementitious material is further increased with an increase in curing time and binder content and filling of the pores between voids. Thus producing a dense structure with a corresponding increase in the stiffness of the soil. The trend of increase the soil stiffness is consisted with [14,15,23,39].

Figure 17 shows the relationship between modulus of elasticity, $E_{50}$, and the unconfined compressive strength. For cement and Petrit T treatments (Figure 17), it is noticed that an increase in soil stiffness is accompanied by an increase in soil strength. The modulus of elasticity can be taken as being between 14 to 24 $q_u$ for Petrit T, and between 16 to 85 $q_u$ for the cement treatment.

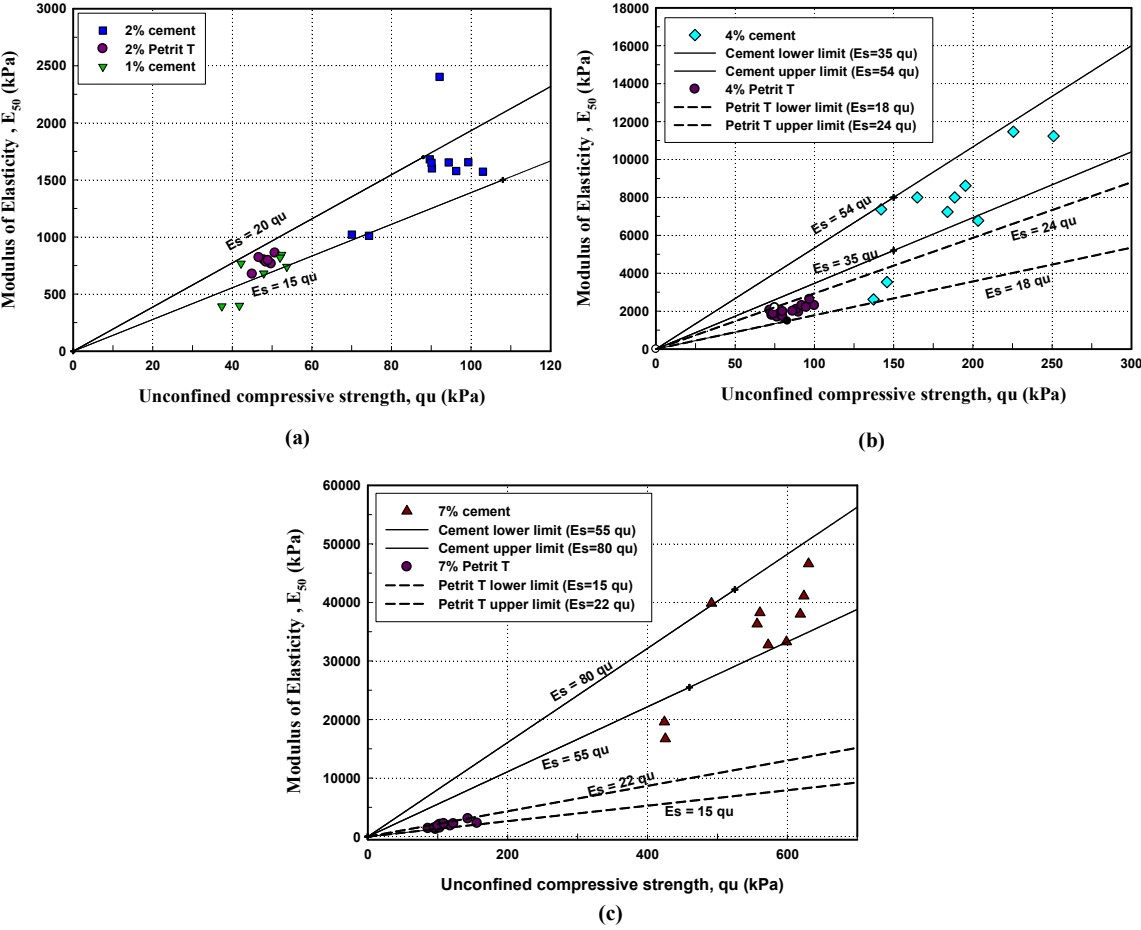

**Figure 17.** Modulus of elasticity versus UCS strength ($q_u$) and binder content for (**a**) treated soil with 1 and 2% cement and 2% Petrit T; (**b**) treated soil with 4% of cement and Petrit T and (**c**) treated soil with 7% of cement and Petrit T.

For a comparison between the effectiveness of cement and Petrit T as binders, an improvement in soil stiffness or strength after treatment can be clarified by the ratio between the stiffness or strength of the treated specimens to the stiffness or strength of the soil specimens without treatment. Table 1 presents the improvement in soil stiffness and strength, values of failure strain, and the relation between $E_{50}$–$q_u$ after treatment with cement or Petrit T. Table 2 shows that a comparable increase in soil stiffness and strength using cement as a binder can be achieved by adding double the amount of Petrit T and using long curing periods. A similar observation is also valid for the reduction in failure strain and the relationship between soil stiffness and UCS.

**Table 2.** Enhancement in soil strength and stiffness, failure strain and the relation between $E_{50}$–$q_u$ after treatment with two binder types.

| Binder Content % | Enhancing in Soil Strength, $q_u$ | | Enhancing in Soil Stiffness, $E_{50}$ | | Failure Strain % | | $E_{50} = q_u$ (Upper and Lower Range) | | | |
|---|---|---|---|---|---|---|---|---|---|---|
| | Cement | Petrit T | Cement | Petrit T | Cement | Petrit T | Cement | | Petrit T | |
| 0 | 1 | 1 | 1 | 1 | 24 | 24 | 6 | 10 | 6 | 10 |
| 1 | 2 | - | 4 | - | 15 | - | 15 | 18 | - | - |
| 2 | 4 | 2 | 9 | 4 | 9 | 14 | 15 | 20 | 15 | 20 |
| 4 | 9 | 4 | 45 | 10 | 5 | 8.5 | 35 | 54 | 18 | 24 |
| 7 | 27 | 6.5 | 180 | 15 | 3 | 7.5 | 55 | 80 | 15 | 22 |

In terms of the relationship between $E_{50}$ and $q_u$, many authors [14,15,18,29,53,55–59] have reached varying conclusions about this relationship. Table 3 presents a comparison between the previous studies and the present study. Moreover, many researchers used different techniques such as using the electrokinetics method and nanomaterials to improve the stability of collapsible soil [60], and developed new procedures and specifications to estimate the mechanical properties of compacted geomaterials for design verification [61]. The results obtained are consistent in general with previous studies.

**Table 3.** Comparison between the relationship of elastic modulus ($E_{50}$) and UCS ($q_u$).

| Material | Upper and Lower Range of Soil Stiffness Times $q_u$ | Reference |
|---|---|---|
| Three type of soil in (silt, silty clay and laterite) treated with cement (7–13%) in Malaysia | $E_{50} = (100-326)\ q_u$ | [15] |
| Swedish clay treated with cement and lime (200 kg/m$^3$ (18–24%)) | $E_{50} = (53-92)\ q_u$ | [59] |
| Clay treated with Cement (3 to 37%) in Finland | $E_{50} = (100-200)\ q_u$ | [53] |
| Bangkok clay treated with cement (5 to 20%) | $E_{50} = (115-150)\ q_u$ | [58] |
| Soft Bangkok clay treated with (10 to 13%) of cement and cement kiln dust with partial replacement of 10 to 20% fly ash | $E_{50} = (99-159)\ q_u$ | [62] |
| Bangkok soft clay at high water content treated with cement (5–35%) and fly ash (5–30%) | $E_{50} = (96-129)\ q_u$ | [57] |
| Chinese marine clay at high salt concentration treated with cement (10 to 20%) | $E_{50} = (150-275)\ q_u$ | [14] |
| Chinese Silt soil carbonated with reactive MgO (5–30%) | $E_{50} = (30-200)\ q_u$ | [55] |
| Marine sediments in France treated with cement, lime and fly ash (3–9%) | $E_{50} = (60-170)\ q_u$ | [56] |
| Swedish sandy clayey silt soil treated with two binder types cement CEM II (1–7%) and by-product Petrtit T (2–7%) | $E_{50} = (16-85)\ q_u$, $E_{50} = (14-24)\ q_u$ | Present study |

## 4. Conclusions

The present study presents a comparative evaluation of the effect of two different binders on the physical and engineering properties of soil after treatment. The study cover both short- and long-term effects and the following conclusions can be drawn:

- Cement is more effective on improving the physical and engineering properties of treated soil. The same effect can be achieved by using the double amount of Petrit T and long curing periods.

- Adding small percentages of the two binder types (up to 10%) has approximately the same trend of behaviors in terms of the decrease the plasticity index. Cement has more effect with further increases in binder content.

- The addition of dry binders of cement and Petrit T have an immediate effect on decreasing the initial water content. In addition, further reduction in soil water content can be observed over time. The reduction in soil water content has a more dominant effect from cement during the first 28 days of curing, compared to Petrit T, which shows gradual decreases in water content over time.

- The density of treated soil with cement and Petrit T is increased with increasing binder content and curing time.

- Liquidity index and the water content to plastic limit ratio are introduced as new indices to define the improvement in the workability of treated soil directly after treatment and over time. The workability improved immediately after treatment with the reduction in the liquidity index. Continuous improvement in soil workability is observed over time due to a further reduction in the liquidity index within the plastic range when the liquidity index (LI) > 0 or when the water/plastic limit ratio is more than one.

- Axial failure strain was decreased as the binder content and curing time increased with greater effect upon cement treatment. Treated soil with 7% cement exhibits a more brittle failure compared to the lower cement content and Petrit T, which shows a more ductile behavior. The failure mode is gradually changed from ductile to brittle failure compared to soil without treatment.

- The pH value of treated soil can provide a useful assessment of the soil–binder reaction. A pH value below 10 is not enough to initialize the pozzolanic reaction and subsequently provides further cementing components over time.

- The particle size distribution curves change towards the granular side by reducing the clay size fraction and increasing the silty size fraction after treatment with a more pronounced effect for cement treatment. This effect is increased with an increase in binder content.

**Author Contributions:** Conceptualization, W.A.-J. and J.L.; methodology, W.A.-J.; validation, W.A.-J., J.L., S.K. and N.A.-A.; formal analysis, W.A.-J.; investigation, W.A.-J.; Visualization, W.A.-J., J.L., S.K. and N.A.-A. Writing—original draft preparation, W.A.-J.; writing—review and editing, J.L.; supervision, J.L., S.K. and N.A.-A.

**Funding:** This research received no external funding.

**Acknowledgments:** The first author thanks the Iraqi Ministry of higher education for offering the scholarships.

**Conflicts of Interest:** The authors declare no conflict of interest.

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
