# Peer review of "A Comparative Evaluation of Cement and By-Product Petrit T in Soil Stabilization"

_applsci, doi:10.3390/app9235238_

Round 1

Reviewer 1 Report

This paper aims to compare and assess the impact of adding low binder amounts of industrial by-product Petrit T associated with cement in order to improve fine-grained soil. The paper is well written and we structured. However, there are some concerns that the authors are suggested to address to improve the overall quality of the the manuscript as follows:

It seems that the discussion of the stabilization methods, despite its significance in the paper, is a bit sketchy. The authors mentioned the advantages of using different stabilizers in this field but failed to address how other researchers and practitioners could improve the stability of the soil by using a variety of different techniques. It is strongly recommended that the authors add a discussion about the bigger picture of different stabilization techniques and showcase what is the state of the art in this field and how this study fits in this picture. This discussion needs to include the latest works in the field;

Hosseini, A., Haeri, S. M., Mahvelati, S., & Fathi, A. (2019). Feasibility of using electrokinetics and nanomaterials to stabilize and improve collapsible soils. Journal of Rock Mechanics and Geotechnical Engineering.

Tirado, C., Rocha, S., Fathi, A., Mazari, M. and Nazarian, S., 2019. Deflection-Based Field Testing for Quality Management of Earthwork (No. FHWA/TX-19/0-6903-1).

Author Response

Author’s Responses to Comments of the Reviewers

We have revised our manuscript in accordance with the reviewers’ comments to improve the manuscript. We are sincerely grateful to the reviewer for the elaborated comments. We respond to review comment or suggestion made in detail below.

Reviewer #1: Comments and author’s response.

Comments #1: This paper aims to compare and assess the impact of adding low binder amounts of industrial byproduct Petrit T associated with cement in order to improve fine-grained soil. The paper is well written and we structured. However, there are some concerns that the authors are suggested to address to improve the overall quality of the manuscript as follows:

It seems that the discussion of the stabilization methods, despite its significance in the paper, is a bit sketchy. The authors mentioned the advantages of using different stabilizer in this field but failed to address how other researchers and practitioner could improve the stability of the soil by using variety of different techniques. It is strongly recommended that the authors add a discussion about the bigger picture of different stabilization techniques and showcase what is the state of the art in this field and how this study fits in this picture. This discussion needs to include the latest works in the field;

Hosseini, A., Haeri, S. M., Mahvelati, S., & Fathi, A. (2019). Feasibility of using electrokinetics and nanomaterials to stabilize and improve collapsible soils. Journal of Rock Mechanics and Geotechnical Engineering.

Tirado, C., Rocha, S., Fathi, A., Mazari, M. and Nazarian, S., 2019. Deflection-Based Field Testing for Quality Management of Earthwork (No. FHWA/TX-19/0-6903-1).

Author’s Responses: Thank you for your comment. The manuscript have been revised and updated, please see line 359-362.

The two suggested references are used also. References 60 and 61.

Reviewer 2 Report

The manuscript titled in 'A comparative evaluation of cement and by-product Petrit T in soil stabilization' described the effect of soil treatment using two different binders, well.

The manuscript is acceptable in Applied Science journal in present form.

Author Response

Thank you very much for your efforts.

Reviewer 3 Report

The paper is well written and clear

Author Response

Thank you very much for your efforts.
